# High-Speed Path Probing Method for Large-Scale Network

**DOI:** 10.3390/s22155650

**Published:** 2022-07-28

**Authors:** Zhihao Luo, Jingju Liu, Guozheng Yang, Yongheng Zhang, Zijun Hang

**Affiliations:** 1College of Electronic Engineering, National University of Defense Technology, Hefei 230037, China; lzh_9226@163.com (Z.L.); jingjul@aliyun.com (J.L.); zhangyongheng_nudt@163.com (Y.Z.); hangzijun17@nudt.edu.cn (Z.H.); 2Cyberspace Security Situation Awareness and Evaluation Key Laboratory of Anhui Province, Hefei 230037, China

**Keywords:** network topology, network path probing, stateless scanning, high-speed probing

## Abstract

In large-scale network topology discovery, due to the complex network structure and dynamic change characteristics, it is always the focus of network topology measurement to obtain as many network paths as possible in a short time. In this paper, we propose a large-scale network path probing approach in order to solve the problems of low probing efficiency and high probing redundancy commonly found in current research. By improving the packet delivery order and the update strategy of time-to-live field values, we redesigned and implemented an efficient large-scale network path probing tool. The experimental results show that the method-derived tool can complete path probing for a sample of 12 million/24 network address segments worldwide within 1 hour, which greatly improves the efficiency of network path probing. Meanwhile, compared to existing methods, the proposed method can reduce the number of packets sent by about 10% with the same number of network addresses found, which effectively reduces probing redundancy and alleviates the network load.

## 1. Introduction

The Internet has a huge scale and complex structure while lacking macrolevel control mechanisms. It has been the research hotspot in network science to discover its network topology from the perspective of network measurement, and it has important implications for our better understanding of large-scale network structures [1,2,3,4]. Network topology discovery refers to obtaining the network structure of the Internet using an active probing approach. According to the granularity of network nodes, network topology discovery can be primarily divided into the IP interface level, router level, PoP (point-of-presence) level, and autonomous domain level [5]. In particular, IP interface level network topology discovery is the basis for network topology discovery at other levels, which is important in order for people to analyze and study network characteristics, assess network security posture, and optimize network structure [6,7,8,9,10,11,12]. In recent years, similar research has emerged in the field of IoT (the internet of things), providing some support for IoT security [13,14].

Large-scale network topology discovery is mainly based on Traceroute active probing technology, which constructs IP packets with specific time-to-live (TTL) field values to obtain the time-out response packets of hop-by-hop router IPs in the network [15,16,17]. The traditional network path measurement approach based on Traceroute mainly uses a synchronous transceiving mechanism, which spends much more time waiting for response packets than the actual transceiving packets, thus severely limiting detection efficiency. Later, the Scamper [18] technique emerged to improve the efficiency of network path probing by simultaneously sending packets through multithreading and was widely used in the Ark project of CAIDA [19]. However, due to the limitation of the number of threads and the large overhead incurred by the frequent creation and destruction of threads, it is still impossible to achieve efficient probing of large-scale networks in a short time.

In recent years, the stateless-based asynchronous packet transceiving mechanism has emerged, which avoids a long time of waiting in synchronous transceiving by decoupling the sending and receiving part of the probing process and achieves a great improvement in the sending rate. On this basis, several tools such as Zmap [20], Yarrp [21], and FlashRoute [22] have been designed and implemented. The probing approach based on the asynchronous transceiving mechanism shows a large improvement in the probing rate compared to the traditional path probing approach, but it does not deal well with probing redundancy. Due to the rate limitation in the network, the large amount of redundancy has some impact on the probing efficiency.

In this paper, we focused on efficient network path obtaining problems in large-scale networks and studied the asynchronous packet transceiving technique based on the state-less scanning mechanism. This paper designed an efficient large-scale network path-prob-ing method by analyzing the advantages of available methods such as Yarrp and FlashRoute and improving the target address sequence generation method and TTL value update strategy. Our contributions include:The design and development of a large-scale network path probing tool. We used the tool to scan the worldwide network at 100 kpps in under an hour and found nearly 600,000 surviving addresses.An analysis of the main factors affecting the probing effect, which provides a basis for a reasonable parameter setting.An analysis of the effect of different probing packages on the probing effect.

The remainder of this paper is organized as follows: Section 2 introduces the existing related work on network path probing; Section 3 describes, in detail, the large-scale network path probing approach proposed in this paper; Section 4 evaluates the tool performance and compares it with state-of-the-art tools (i.e., Yarrp, FlashRoute); and finally, Section 5 concludes the main achievements of this paper and discusses potential future research directions.

## 2. Related Work

In large-scale network probing, research has focused on both reducing the probing redundancy and improving the probing efficiency.

In terms of reducing probing redundancy, Donnet et al. [23] proposed the double-tree algorithm. Depending on the difference between the probing source and the probing target, the algorithm categorizes the probing results into a source tree with the source as the root and a target tree with the target as the root, aiming to reduce the probing redundancy of each tree. The algorithm assigns a local stop set and a global stop set to the source, respectively, and targets trees for storing the discovered addresses. When an address in the stop set is found, the probing can be stopped in time to avoid multiple probing of the same address. To make full use of the algorithm effect, the probing order is different from the traditional Traceroute. Each probing starts from the h-hop in the middle of the path and extends the probing in two directions, respectively. In addition, the probe does not continue forward or backward when it discovers an address in the stop set. This approach can effectively reduce the probing redundancy; however, it also suffers from the difficulty of determining the value of h and a large amount of communication between the probing source and the control node.

In terms of improving the probing efficiency, Zmap [20] decouples the packet sending and receiving modules of probing for the first time and performs asynchronous port probing on the target network through the stateless mechanism. With the support of hardware devices, the packet sending rate can reach 10 million per second, which enables the probing of the entire Internet range in a short time. Meanwhile, the stateless scanning mechanism also causes problems such as traffic concentration and response identification. On the one hand, high-speed probing generates huge, concentrated traffic, which can easily cause routing overload. On the other hand, the stateless mechanism means not recording status information such as the destination address and port number of each probe, so it is necessary to verify the packets after receiving them and distinguish between probing response packets and background traffic packets. To deal with the above two problems, Zmap corresponds by designing probing packets and randomizing the order of sending packets.

Based on the stateless scanning mechanism of Zmap, Robert et al. [21] designed and implemented the network path probing tool Yarrp, which applies asynchronous transceiving to the field of network path probing and greatly improves the efficiency of path probing. Yarrp uses randomization of the target address and TTL value pairs instead of randomization of the target address in Zmap. Yarrp limits the probing range and guarantees the integrity of the path probing by setting the maximum TTL value for the target. However, the redundancy that exists in the path probing is not handled, which results in a large number of invalid probing packets in the network path probing.

Due to the ICMP rate limitation problem of routers, Huang et al. [22] improved Yarrp from the perspective of reducing the probing redundancy and developed the path probing tool FlashRoute. FlashRoute combines the double-tree algorithm with asynchronous network path probing, dividing the probing process into two phases: the pre-probing phase and the probing phase. The pre-probing phase is used to determine the hop count of the target address. The probing phase starts backward probing for each target from the maximum number of hops, stores the probing results in the stop set, and stops probing in time at repeated addresses to reduce probing redundancy.

Although available large-scale network path probing methods use the stateless scanning mechanism to improve path probing efficiency, there are certain weaknesses in the probing redundancy and decentralized processing of probing packets. Specifically, there are three main problems.

Firstly, Yarrp randomizes the target address and TTL value pairs, which has a better effect on probing packet decentralization but does not consider the redundancy in the path probing process, which results in a large number of invalid probing packets.

Secondly, FlashRoute can stop duplicate probes in time by introducing the stop set in the double-tree algorithm, which reduces the number of backward probes and decreases the probe redundancy in path probing. However, there is the problem that the TTL values between the target addresses are too concentrated, which can easily lead to a specific hop number of routers reaching the ICMP rate limit, resulting in target response loss and affecting the path detection effect.

In addition, these approaches both construct the target address sequence by employ-ing pseudorandom numbers, which can guarantee the address distribution to be dispersed over time but cannot make sure that the addresses uniformly distribute across network prefixes.

## 3. Approach to Design

In this paper, we designed an asynchronous probing method for large-scale network path probing based on the stateless scanning mechanism and constructed probing packets based on the identification field of the IP packet and the identification field of the ICMP packet. To deal with the problem of huge transient traffic in asynchronous probing, we proposed a target address sequence generation approach based on inverse binary sequences, which meant the target address prefixes could be evenly distributed to disperse the traffic. To prevent triggering the ICMP rate limit of the router, this paper designed a bidirectional path probing approach based on the dynamic update policy of TTL values, which keeps the TTL value from becoming too concentrated and causing the response rate to decrease.

### 3.1. Probing Packet Construction Approach Based on the Identification Field of the IP Packet and Identification Field of the ICMP Packet

The key to probing packet construction is to save the status information of the prob-ing in the probing packet so that the probing information can be verified and recovered when receiving the response packet. The packet type can be ICMP, UDP, or TCP, etc. This paper introduced the probe packet design with the ICMP request packet as an example as shown in Figure 1.

The probing information to be saved mainly includes the TTL value of the probing and the hash value of the target address for verification. This paper used the identification field of the IP packet to store the TTL value and the identification field of the ICMP packet to store the hash value of the destination address. Similarly, the TOS field of the IP packet and the serial number field of the ICMP packet could also be used for the storage of status information. When receiving a response packet, we first verified the hash value of the target address and filtered the packets with incorrect hash values. Regarding the packets that passed the validation, we further extracted the probe information such as response address and TTL value to update the stop set and address information block.

### 3.2. Destination Address Sequence Generation Approach Based on the Inverse Binary Sequence

High-speed asynchronous probing results in huge probing traffic, and the concentrated traffic tends to overload the target network if network path probing is performed in the normal order. FlashRoute generates a pseudorandom destination address sequence with multiplicative groups, which alleviates the problem of concentrated traffic to some extent, but it does not take the characteristics of real network routing into account.

In real networks, routers mostly adopt classless interdomain routing (CIDR) [23]. Routers tend to aggregate multiple neighboring network segments in the same direction and choose routes using the longest prefix matching [24]. Therefore, network traffic is also aggregated by the destination address prefix, so the destination address sequence needs to be distributed as evenly as possible across network address prefixes. This paper generated the destination address sequence for a specific network segment with the inverse binary sequence to ensure that the minimum distance between the same prefix addresses in the sequence was maximized.

The inverse binary sequence refers to the new sequence generated by inverting the normal binary sequence as shown in Figure 2. It can make the distance between any adjacent same prefix address constant while ensuring that the minimum distance between the same prefix address is maximized, which is equal to the distance between any adjacent same prefix address as $2^{n}$ (*n* is the prefix length).

### 3.3. Bidirectional Path Probing Approach Based on the TTL Value Dynamic Update Strategy

ICMP rate limits are commonly present in routers [25], and most routers can only send 500 or even fewer ICMP response packets per second. Therefore, when probing multiple targets, the TTL values need to be scattered to avoid the loss of response packets caused by the ICMP rate limit of the trigger hop router. Yarrp limits the probing scope by setting the maximum TTL value and generates the probing sequence by randomizing the target address and TTL value pairs, which can effectively alleviate the network load; however, it needs to probe the complete target address and TTL value space with large probing redundancy. FlashRoute determines the hop count of the target address by pre-probing, which narrows the scope of TTL values. Each target address is probed in the decreasing order of TTL values, while the stop set is used to reduce probing redundancy. However, the TTL values tend to close in each round of probing, which can result in a large load on a particular hop router.

In this paper, we started with a random initial TTL value for each target address, carried out path probing in the order of backward probing, and then forward probing on the whole. During the probing process, the TTL value was adjusted following the dynamic update strategy. On the one hand, TTL values could be spread out as much as possible, and on the other hand, probing redundancy could be reduced by the stop set. To generate the stop set properly, this paper used multi-round probing, where each round probing was performed for all target addresses in one hop and waited for a while after each round probing to make sure that most of the response packets were received, and the stop set was updated in time. According to the probing phase and the response packet, the probing status–transition relationship is shown in Figure 3.

To explain in detail the strategy proposed in this section, we use the following definitions:initTTL, initial TTL value, which means the TTL value of the initial probing for a certain destination.curTTL, current TTL value, which means the TTL value of the current probing for a certain destination.maxTTL, maximum TTL value, which means the maximum TTL value of the probing for a certain destination.rspTTL, response TTL value, which means the TTL value in the response packet.

During the probing, the TTL value needed to be adjusted in time according to the probing phase and response packets to reduce the probing redundancy. According to the different probing phases, the TTL value update policy was divided into two sections: the sending phase and the receiving phase. Table 1 shows the update policy at sending phase.

The update at sending phase was mainly the natural update of curTTL after sending a specific probe packet. Firstly, it judged whether the probing direction was backward or forward, and secondly, it judged whether the probing in that direction was finished. For example, when curTTL did not exceed initTTL, the probing direction was backward, and curTTL only needed to be minus 1 in general. However, when curTTL was 1, it meant that backward probing was finished, and curTTL needed to be set to initTTL+1 to enter the forward probing phases. Similarly, when curTTL was greater than initTTL, the probing direction was forward, and curTTL needed only to be added 1 in general. However, when curTTL was maxTTL, it meant that the forward probing was finished, which also meant that the whole probing was finished, and the destination address needed to be removed from the address information block linked list. Table 2 shows the update policy at receiving phase.

The receiving phase update was mainly based on the response packets and the stop set. Firstly, it judged the response message type. Secondly, it judged whether the response address was in the stop set, and finally, it updated the address information block according to the probe direction. For example, in the case of TTL unreachable packets, we first judged whether the response address was in the stop set, and directly added the response address to the stop set if it was out of the stop set. However, it was necessary to judge the probing direction further. If the response TTL value did not exceed initTTL, it meant that the probing phase was in the backward probing phase, and the curTTL needed to be set to initTTL+1 to turn from backward probing to forward probing. If the response TTL value was greater than initTTL, the probing phase was in the forward probing phase, and initTTL needed to be reset to restart the path probing. In the case of ICMP reply packets, we only needed to judge the probing direction. If the response TTL did not exceed initTTL, the probing phase was a backward probing phase, and we needed to update maxTTL to the response TTL value, reset initTTL, and restart the path probing. If the response TTL value was greater than initTTL, the probing phase was the forward probing phase. The whole probing was finished as normal, and the destination address needed to be removed from the address information block linked list.

### 3.4. Design of Destination Storage Structure

Based on FlashRoute [22], we used a similar data structure to store the state information of the probing process in this paper as shown in Figure 4. We optimized the time complexity of the algorithm using the combination of a cyclic double-linked list and array to store the destination address information. There are three main reasons for this. Firstly, the use of a cyclic double-linked list is convenient for finding the next address to be probed from the current probe address. Secondly, using arrays can be faster to query information about the given destination address so that it can be updated during the probing. Thirdly, since the double-tree algorithm needs to remove the probing destination by the stop set, we used the array for positioning and the double-linked list for deletion to remove the address with lower complexity.

We used a structure called address information block to store information about each destination address. The address information block mainly contained probing-related information, such as the destination address, the cyclic double-linked list pointer, the next TTL value to be probed, the probe path, etc.

### 3.5. Algorithm Description

To implement asynchronous probing based on a stateless scanning mechanism, we needed to decouple the functions of both the sending and receiving of messages, so the algorithm was divided into two parts: a packet sending algorithm and a packet receiving algorithm.

The packet sending algorithm started with sampling the input destination address segments by /24 network segments and generating the initial linked list of address information blocks using the inverse binary sequence. Next, we launched multiple rounds of probing on the linked list and kept a certain waiting time after each round of probing to receive the response packets in time so that the address information blocks could be dynamically updated in time to reduce the probing redundancy. Algorithm 1 is described as follows:
**Algorithm 1** Packet Sending Algorithm **Input:** Destination address segment DesAddrSeg, Waiting time T. **Output:** Part of the statistical information of the probing (the number of sending packages, the time of probing).1 **Function** SendThread(DesAddrSeg, T):2 SamAddr ← DesAddrSegs.Sample()3 AIBList ← **new** AIBList(SamAddr)4 SendQueue ← **new** SendQueue()5 SendCount ← 06 StartTime ← Time.Now()7 **while not** AIBList.empty() **do**8   **for** AIB in AIBList **do**9      packet ← GeneratePacket(AIB)10     SendQueue.Add(packet) 11     AIB.Update() 12     SendCount ← SendCount + 113   **end for**14   SendQueue.Transmit()15   Wait(T)16 **end while**17 EndTime ← Time.Now()18 ScanTime ← EndTime—StartTime19 **return**20 **end Function**21 22 **Function** Sample(**this** AddrSegs): 23 **for** AddrSeg **in** AddrSegs:24   Addr ← random_address(AddrSeg) 25   SamAddr.Add(Addr) 26 **end for**27 **return** SamAddr28 **end Function**29 30 **Function** GeneratePacket(AIB):31  packet.IP.SrcAddr ← LocalIP32   packet.IP.DstAddr ← AIB.Addr33  packet.IP.Id ← AIB.curTTL 34  packet.IP.TTL ← AIB.curTTL35   packet.ICMP.Id ← Hash(AIB.Addr)36  **return** packet37 **end Function**38 39 **Function** Update(**this** AIB):40  **if** AIB.curTTL == 1: 41    AIB.curTTL ← AIB.initTTL + 1 42  **else if** AIB.curTTL == maxTTL: 43    AIBList.Remove(AIB)44  **else if** AIB.curTTL ≤ AIB.initTTL: 45    AIB.curTTL ← AIB.curTTL—146  **else**:47    AIB.curTTL ← AIB.curTTL + 1 48  **end if**49  **return**
50 **end Function**

In the description of the packet sending algorithm, the SendThread function described the whole process of the sending part. In addition, we also used three key functions: Sample, GeneratePacket, and Update. Within the Sample function, we sampled the destination address and generated the address sequence based on the inverse binary sequence, which is described in detail in Section 3.2. Within the GeneratePacket function, we used the method described in Section 3.1 to populate each field of the probing packet so that the packet could be verified when the response packet was received. Within the Update function, we updated the TTL value of the target address using the dynamic update policy proposed in Section 3.3, which is similarly described in Table 1. A detailed UML representation of the algorithm is provided in Figure 5.

The packet receiving algorithm constantly received probing response packets and verified them, then updated the address information blocks and the stop set according to the TTL value dynamic update policy. The algorithm set a timeout time to monitor the completion of the probing and finally, produced the probing result. Algorithm 2 is described as follows:
**Algorithm 2** Packet Receiving Algorithm **Input:** Timeout time T.
 **Output:** Address information block array, part of the statistical information of the probing (the number of receiving packages, the number of discovered addresses).1  **Function** RecvThread(T):2  RecvQueue ← **new** RecvQueue() 3  KeepListen() 4  RecvCount ← 0 5  AddrSet ← **new** Set()6  **while** not RecvQueue.empty(T) **do**
7    packet ← RecvQueue.pop()8    **if not** Check(packet) **do**9      **continue**
10   **end if**
11   Info ← packet.Resolve() 12   AIBList.Update(Info) 13   AddrSet.Add(Info.Addr) 14   RecvCount ← RecvCount + 115 **end while**16 **return**17 **end Function**18 19 **Function** Check(packet): 20 **if not** packet.IP.DstAddr == LocalAddr **do**21   **return false**22 **else if not** Hash(packet.IP.SrcAddr) == packet.ICMP.Id **do**23   **return false**24 **else do**25   **return true**
26 **end if**
27 **end Function**
28 29 **Function** Resolve(**this** packet): 30 Info.Addr ← packet.IP.SrcAddr 31 Info.TTL ← packet.IP.Id 32 Info.DstAddr ← packet.ICMP.Data.IP.DstAddr 33 **return** Info 34 **end Function**3536 **Function** Update(**this** AIBList, Info): 37 AIBList[Info.DstAddr].Route[Info.TTL] ← Info.Addr 38 **return**
39 **end Function**

Similar to the packet sending part, the whole process of the packet receiving algorithm was concentrated on the RecvThread function. The Check function was mainly used for the verification of the received packets, filtering a large number of response packets that were not generated by the probing. The Resolve function was used to parse the response packets that passed verification and extract the valuable path information from them, while the Update function was used to further update the results into the AIBList. A detailed UML representation of the algorithm is provided in Figure 6.

We performed a formal verification of the algorithm as a way to theoretically illustrate the validity of the algorithm. The validation of the algorithm mainly consisted of correctness, terminability, and high time efficiency. Firstly, the correctness of the algorithm was guaranteed using the network protocol. For each probe, either no reply packets existed, or a normal reply was made according to the network protocol. Secondly, the terminability of the algorithm was determined by the overall design of the algorithm. For the packet sending algorithm, there was an upper limit to the number of cycles, so it was bound to terminate. For the packet receiving algorithm, we set a timeout to ensure that the algorithm could be terminated. Finally, for the time efficiency of the algorithm, we present the following definitions or assumptions:$N_{t}$, the maximum number of threads, set to 1000.$T_{r}$, the average reply time considering the nonresponse case, set to 1 s.$S_{NIC}$, the maximum network interface card (NIC) packet sending rate, determined by the hardware device, set to 10,000,000 packets per second.$B_{w}$, bandwidth, determined by the network environment, set to 100 Mb per second.
$P_{s}$, the probe packet size, determined by the probe packet construction, set to 100 bits.

The probing approach based on the stateless scanning mechanism decoupled sending and receiving packets, and the NIC used full-duplex mode, so sending and receiving packets did not affect each other. The sending rate was jointly determined by the bandwidth and the maximum NIC sending rate, which could be reached at 1 Mpps.
$$S=min\left(S_{NIC},S_{b}\right)=min\left(S_{NIC},\frac{B_{w}}{P_{s}}\right)$$

Compared to this, based on the traditional synchronous sending and receiving mechanism, the maximum packet sending rate could be reached at 1 kpps.
$$S=min\left(S_{t},S_{b}\right)=min\left(\frac{N_{t}}{T_{r}},\frac{B_{w}}{P_{s}}\right)$$

Theoretically, there was a great improvement in time efficiency compared to the traditional synchronous sending and receiving mechanism. In order to avoid negative effects on the network, we used the maximum packet sending rate of 100 kpps in the experiments in Section 4, and the experimental effect can also be corroborated with the formal verification.

Another factor that affects the time efficiency lies in the response ratio of destination addresses, i.e., the ratio of probes with destination responses to the total number of probes. Here, we also present some relevant definitions and assumptions:$T_{p}$, the processing interval time of the destination address, considered as a constant.
$T_{a}$, the average sending interval time of the probing source, considered as a constant.$T_{s}$, the sending interval time of the probing source.$R_{r}$, the ratio of probes with destination responses to the total number of probes.The case of anonymous routers was not considered, i.e., the destination address certainly responded to the probing packet as long as it satisfied the processing interval time requirement of the destination address.To maintain a high response ratio, the average sending interval time was greater than the processing interval time of the destination address i.e., $T_{a}>T_{p}.$


Our approach used the inverse binary sequence to generate the target address sequence, and the distance between any adjacent same prefix address was constant, so it can be assumed that the sending interval time was constant and equal to the average sending interval time.
$$T_{s}=T_{a}>T_{p}$$

Since the sending interval time was greater than the processing time, the destination address always responded to the probing packet.
$$R_{r}=1$$

The Yarrp and FlashRoute approaches use the pseudorandom sequence to generate the destination address sequence, so the interval between adjacent addresses with the same prefix was random and may be assumed to conform to a normal distribution.
$$T_{s}\sim N\left(T_{a},\sigma^{2}\right)$$

Therefore, the response ratio was equal to the probability that the sending interval time exceeds the processing interval time.
$$R_{r}=P\left(T_{s}>T_{p}\right)=1-F\left(T_{p};T_{a},\sigma \right)$$

It was obvious that this response ratio was less than 1. The analysis shows that our method has a higher response ratio than Yarrp and FlashRoute, which is also shown in the experiments in Section 4.2.4. In formal probing, the presence of anonymous routers, the response policy of the target address, and the network environment of the experiment all have a significant impact on the results, so a high response ratio was not observed in Section 4.2.4.

## 4. Performance Evaluation

### 4.1. Tool Design and Implementation

Most of the available network path probing tools are based on the Linux platform, and there is a vacancy in the network path probing tools based on the Windows platform. This paper implemented a large-scale network path probing tool based on the .Net platform and C# language by Winpcap and validates its legitimacy.

The tool adopted a modular design, consisting of eight modules: input, address generation, packet construction, packet sending, packet receiving, packet verification, packet handling, and output. These modules were independent of each other and cooperated to implement the path probing approach based on the stateless mechanism, as shown in Figure 7.

The input module was mainly used for the input of the destination address set, removing private addresses from the destination address set resulting in the actual destination address to be probed. The address generation module mainly sampled addresses for the destination address set according to the /24 network segment and initialized the address information block linked list in the order of the inverse binary sequence. The packet construction module was used to take out the destination address and TTL value from the address information block linked list, construct the probing packet according to the different probing packet types, and add it to the send queue. The packet sending module was used to asynchronously send probing packets from the send queue and to update the corresponding address information blocks. The packet receiving module was used for receiving response packets and adding them to the receive queue. The packet validation module removed the response packets for verification from the receive queue and filtered the packets failing verification. The packet handling module extracted the path probing results from the response packets according to the packet format and updated the corresponding address information blocks. The output module was mainly used for outputting and storing the probing results.

### 4.2. Tool Validation and Analysis of Experimental Results

#### 4.2.1. Coverage of the Test Network

We chose ChinaNet (AS4134) of China Telecom as the test network because it is one of the largest autonomous domain networks in China, with more than one-third of the total number of network addresses in China. We evaluated the various performances of the tool through the AS4134 network, and moreover, we conducted similar experiments on the global Internet in order to check the scalability of the tool. The network data used in the experiment were obtained from the BGP routing table (26 March 2022) of the public project RouteViews [26] at the University of Oregon, and through the analysis, we ascertained that AS4134 contained 771 network routable address prefixes, and 436,772 destination addresses after sampling according to /24 network segmentation. The experiment design focused on the possible factors affecting the probing effect and evaluated the effect of the experiment based on the number of sending packets, probing time, the number of received packets, and the number of discovered addresses during the probing. Finally, we compared it to available large-scale path probing tools to verify the validity of the approach.

#### 4.2.2. Possible Factors Affecting the Probing Effect

Through the analysis of available path probing approaches and Winpcap, this paper concludes that three factors may affect the probing effect: sending rate, maxTTL, and send queue size. In this paper, we analyzed the effect of these three factors on the probing effect of our tool through experiments.

In studying the impact of each factor on the probing effect, we controlled the other factors to be consistent and set to a relatively reasonable value. For example, when analyzing the impact of sending rate on the experimental effect, we set the maxTTL of the tool to 20 and the sending queue size to 1024. We further analyzed the impact of sending rate on the probing effect by adjusting the sending rate to observe the changes in the indicators.

##### Sending Rate

As seen in Figure 8, the probing time is 465 s at the sending rate of 10,000 packets per second. As the sending rate increases, the probing efficiency significantly improves, and the probing time decreases to 102 s when the sending rate reaches 100,000 packets per second. Increasing the packet sending rate does not strictly reduce the probing time in the inverse ratio because of the waiting and handling time between each round of probing, but the probing time can maintain a decreasing trend in general. From the perspective of the number of sending packets, as the sending rate increases, the number of sending packets also increases slightly, as the faster sending rate may cause the response packets not to be received in time, delaying the TTL value update and generating some probing redundancy.

Figure 9 presents the variation of receiving packets with the sending rate. It can be seen that as the sending rate increases, the target network load increases, the number of response packages decreases, and the number of receiving packages also slightly decreases. In terms of the number of discovered addresses, the total number varies little with the sending rate and stays around 37,000. In other words, when carrying out path probing at a high packet rate, we can also ensure the integrity and effectiveness of probing; therefore, the actual probing can reduce the probing time by appropriately increasing the packet rate.

##### maxTTL

Figure 10 shows the variation of the number of sending packets and probing time with maxTTL. It can be seen that with the same sending rate, the probing time and the number of sending packets keep the same trend. With the increase in maxTTL, the number of sending packets and probing time linearly increase. The main reason for this is that the increase in maxTTL extends the range of TTL values and the probing space, which results in more sending packets and probing time.

As shown in Figure 11, the number of discovered addresses is at a threshold of a maxTTL of 20 where the number of discovered addresses keeps increasing until it reaches the maxTTL and remains stable after the maxTTL is met. To a certain extent, this can also illustrate the small-world characteristic of the Internet: in a large-scale network, different addresses may connect in only a short number of hops. After the maxTTL exceeds the threshold value, no more addresses will be found by increasing maxTTL. However, the number of receiving packets maintains a constant increase during the process because the increase in sending packets brings more probing redundancy, resulting in a growing number of response packets.

##### Send Queue Size

According to Figure 12, the number of sending packets and probing time does not vary much with the send queue size when the send queue is small, but there is a rapid increase when the send queue size reaches or exceeds 215. Combined with Figure 13, the number of receiving packets and the number of discovered addresses have a similar variation. It can be seen that an oversized send queue will have a direct impact on the probing results, not only resulting in a higher number of sending packets and probing time but also in a loss of probing effect. This occurs as an oversized send queue may cause congestion of the probing packets in routing and trigger the ICMP rate limit of the router. As a result, fewer response packets are received so more probing destinations cannot stop probing in time, which results in more probing costs.

#### 4.2.3. Impact of Different Probing Packets on the Probing Effect

The phenomenon of message filtering is prevalent in the network, and different types of packets are filtered somewhat differently in the network. This is very important for the integrity of path probing. To study the impact of probe packets on the path probing effect, we conducted path probing experiments on the target network using three common probe packets, ICMP Request, UDP, and TCP SYN, respectively. We implemented the path probing experiments on AS4134 at a 100 kpps sending rate, maxTTL of 20, and send queue size of 1024. The experimental results are shown in Table 3.

As shown in Table 3, there is some difference in the probing effect of the different probing packets, but this difference is not very obvious. In general, ICMP and UDP packets have a relatively close probing effect, while TCP SYN packets have a slightly worse effect. To some extent, this could indicate that there may be more stringent filtering of TCP packets, especially TCP SYN packets in the network. In contrast with ICMP and UDP packets, TCP SYN packets are mainly used in the network to establish TCP connections and occupy target host resources, which may also lead to strict filtering of TCP SYN probe packets by firewalls. For different probing targets, we can combine several probing packets for path probing, and at last, the results from probing can complement each other to form a more integrated outcome.

#### 4.2.4. Comparison with Available Tools

In this paper, we implemented the probing tool with ICMP probing packets at a 100 kpps sending rate, maxTTL of 20, and the send queue size of 1024 to probe AS4134. We adopted the same configuration for the available large-scale path probing tools Yarrp and FlashRoute, and we consistently used ICMP Request probing packets to test the target network in our experiments. The results of comparing the experimental results of Yarrp, FlashRoute and ours are shown in Table 4.

The experiment results show that our tool can greatly reduce the number of sending packets compared to available path probing tools. Our tool is about 10% lower than FlashRoute and close to 50% lower than Yarrp, and the probing time has also been significantly reduced. Meanwhile, our tool has the highest receiving and sending ratio with the least number of sending packets, and the number of discovered addresses remains consistent with available tools. It shows that our approach can reduce the probing redundancy as much as possible and effectively reduce the probing costs while guaranteeing the probing effect.

In order to observe the extensibility of the method, we also carried out a path probing experiment using different probing packets on 930,000 routable prefixes worldwide (12 million/24 network segments in total). The experiment results show that our approach can achieve path probing on the sample addresses of all /24 network segments worldwide within 1 h.

The number of discovered addresses varies significantly for different probing packets. We found 597,070 surviving addresses using ICMP probe packets, while 532,168 and 535,698 surviving addresses were found using UDP probe packets and TCP SYN probe packets. In contrast with our observation in AS4134, this phenomenon shows that there are also differences in packet filtering features in different network environments. For complex network environments, we can reduce the impact of packet filtering on the probing results by using multiple probing packets.

#### 4.2.5. Impact on the Network

Higher probing rates may impose some burden on the target network and can easily be mistaken as attacks. We followed the recommended guidelines for good Internet citizenship provided in [27] to mitigate the potential impact of our probing.

Firstly, we generated the destination address sequence based on the inverse binary sequence to avoid a focused scan of the partial network as much as possible. Secondly, we employed a TTL value dynamic update strategy to reduce probing redundancy and tried our best to avoid target-specific overload. In addition, we used the ICMP request packet to probe the target network, which is essentially similar to ping, avoiding the additional cost of establishing port connections. Unfortunately, different probing packets may reach the same destination address in network path probing, so redundancy cannot be completely avoided.

Due to the tree structure of the probing results, the average number of probes was smaller far from the probing source with a large address base and larger near the probing source with a small address base. In addition, the TTL value dynamic update strategy can reduce the redundancy of proximal probes to some extent.

In the AS4134 path probing experiment, we sent 4,608,352 probing packets, received 1,565,637 response packets, and found 37,853 surviving addresses with the probing time of 102.717 seconds. From the responses received, 2.7% of the destination addresses received more than 100 probing packets, while the majority of the destination addresses received just less than 10 probing packets.

Similarly, in the global path probing experiment, we sent 137,963,354 probing packets, received 36,388,785 response packets, and found 597,070 surviving addresses with the probing time of 3,244.311 seconds. From the responses received, 0.3% of the destination addresses received more than 1000 probe packets, while the majority of the destination addresses received just less than 100 probing packets.

## 5. Conclusions

In this paper, we propose a large-scale network path probing approach based on a stateless mechanism to overcome the problems of low efficiency and high redundancy in traditional path probing approaches. Our approach effectively improves the path probing efficiency through asynchronous transceiving, reduces the probing redundancy with the double-tree algorithm, and makes full use of available probing results to optimize further probing. Compared to available large-scale path probing approaches, our approach can effectively reduce probing redundancy and further improve the path discovery efficiency in large-scale networks with high probing integrity.

The Internet has a complex network topology. On the one hand, the topology obtained from a single probing source is not comprehensive [28,29,30,31]. On the other hand, reducing the probing redundancy often means reducing the discovery of additional routes because of the load balancing phenomenon [32]. The distributed architecture [33] will help to improve the integrity and reliability of network topology discovery and can support router identification on the basis of this. How to introduce the approach into multi-source topology probing and strike a balance between reducing probing redundancy and discovering additional routes will be investigated in future work.

## Figures and Tables

**Figure 1 sensors-22-05650-f001:**
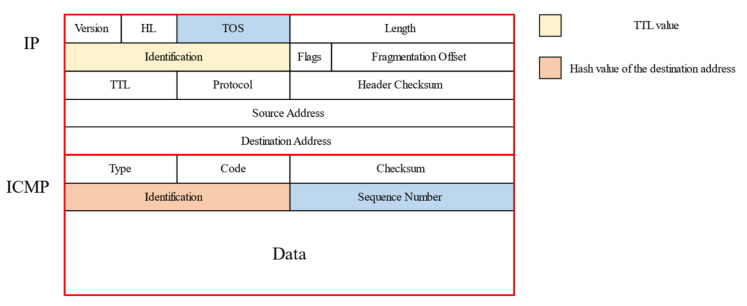
The structure of the probing packet based on the identification field of the IP packet and the identification field of the ICMP packet.

**Figure 2 sensors-22-05650-f002:**
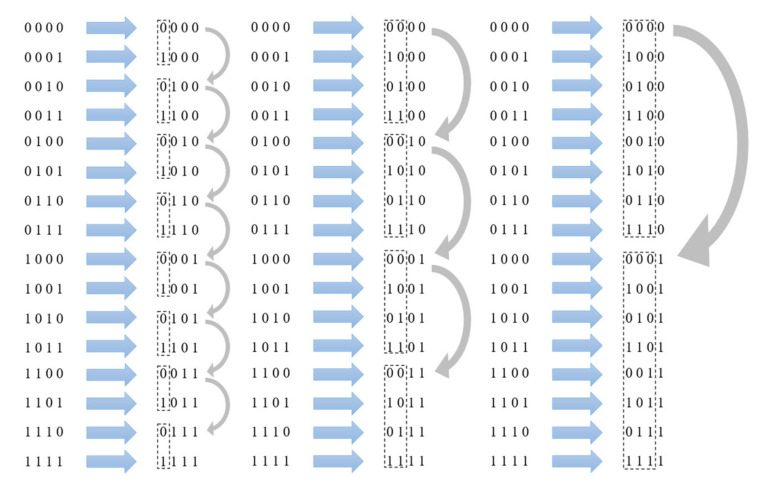
Inverse binary sequence.

**Figure 3 sensors-22-05650-f003:**
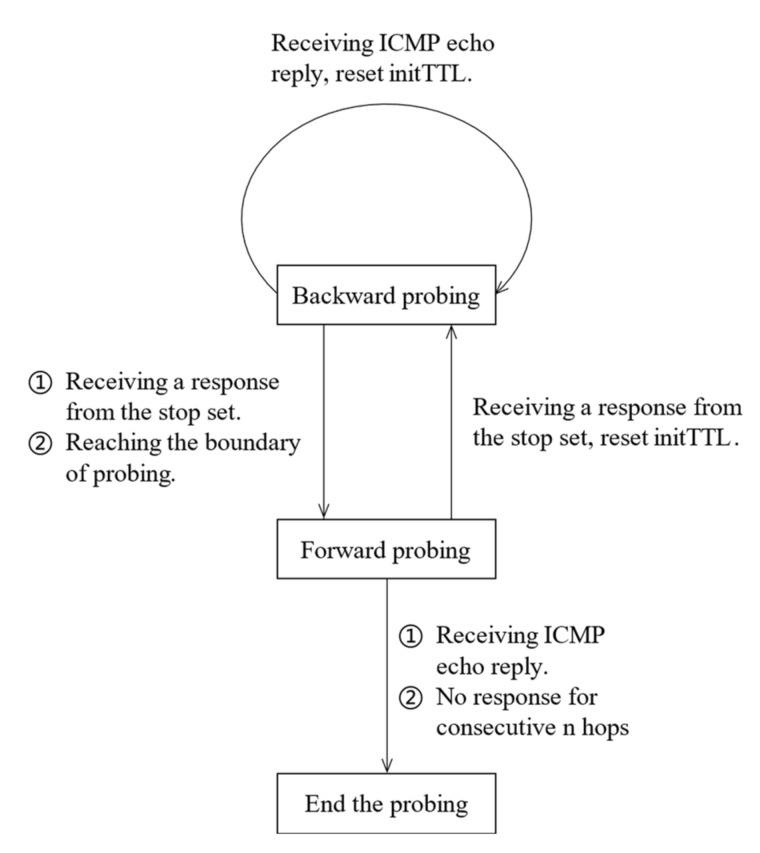
Probing state–transition relationship.

**Figure 4 sensors-22-05650-f004:**
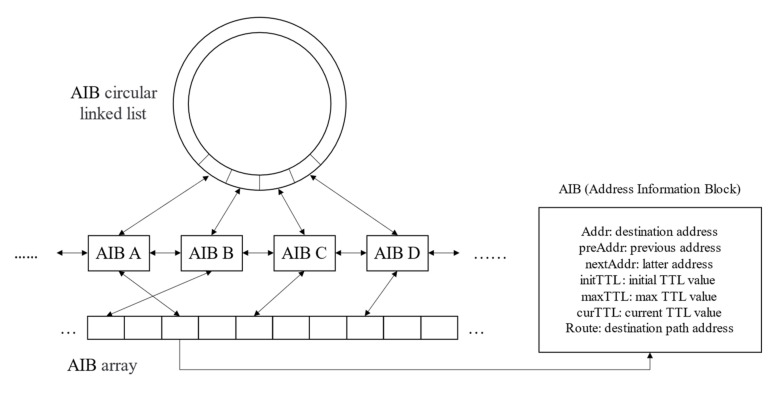
Destination storage structure.

**Figure 5 sensors-22-05650-f005:**
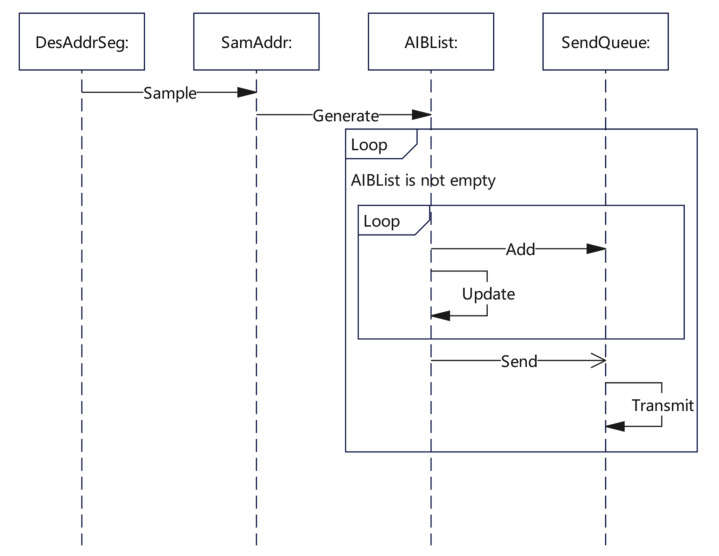
Packet sending algorithm Sequence Diagram.

**Figure 6 sensors-22-05650-f006:**
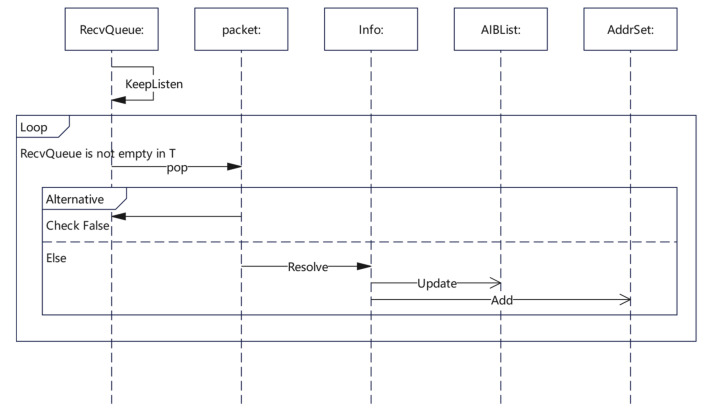
Packet Receiving algorithm Sequence Diagram.

**Figure 7 sensors-22-05650-f007:**
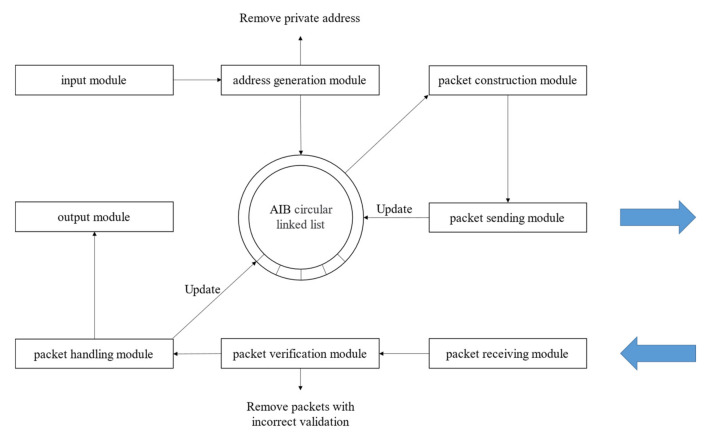
Functional module design.

**Figure 8 sensors-22-05650-f008:**
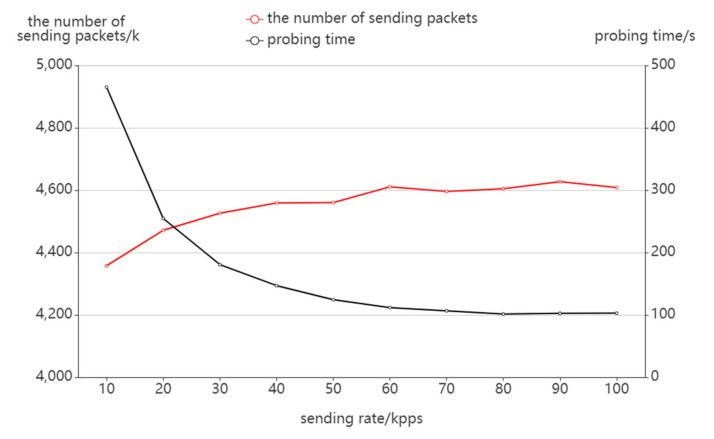
The effect of sending rate on packet sending.

**Figure 9 sensors-22-05650-f009:**
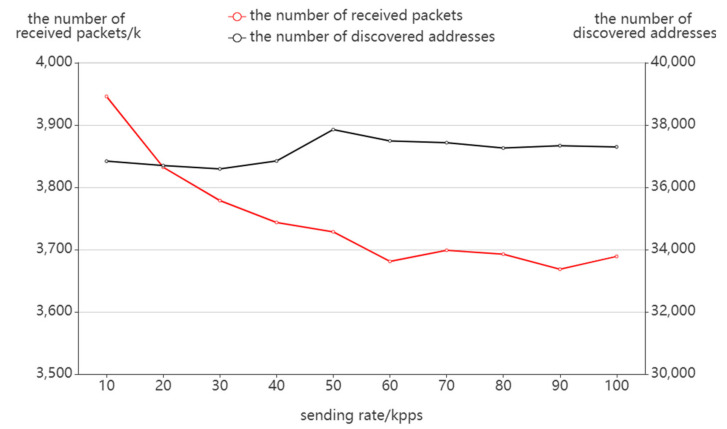
The effect of sending rate on packet receiving.

**Figure 10 sensors-22-05650-f010:**
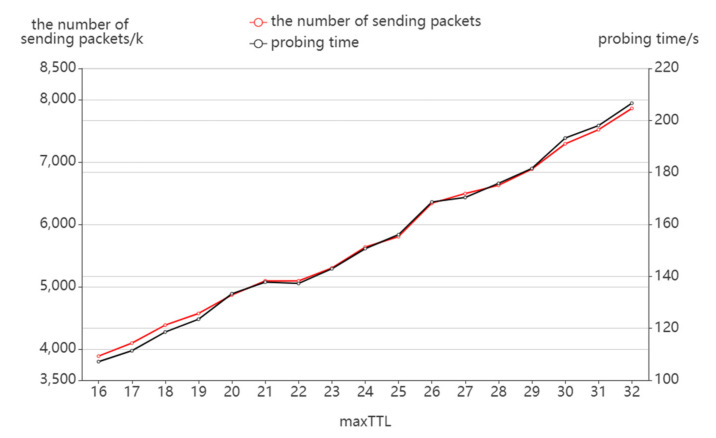
The effect of maxTTL on packet sending.

**Figure 11 sensors-22-05650-f011:**
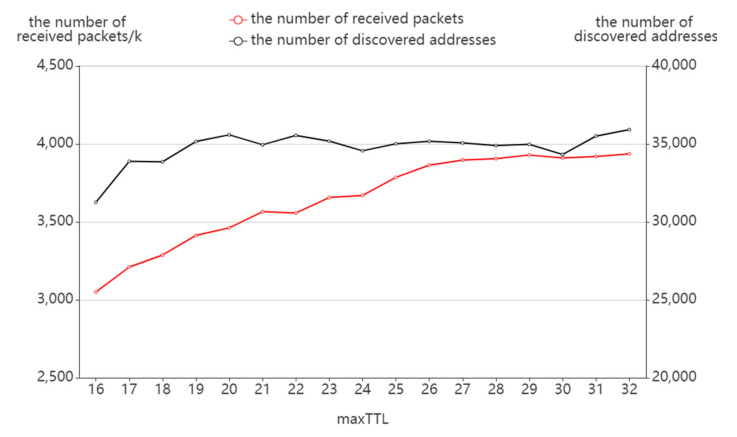
The effect of maxTTL on packet receiving.

**Figure 12 sensors-22-05650-f012:**
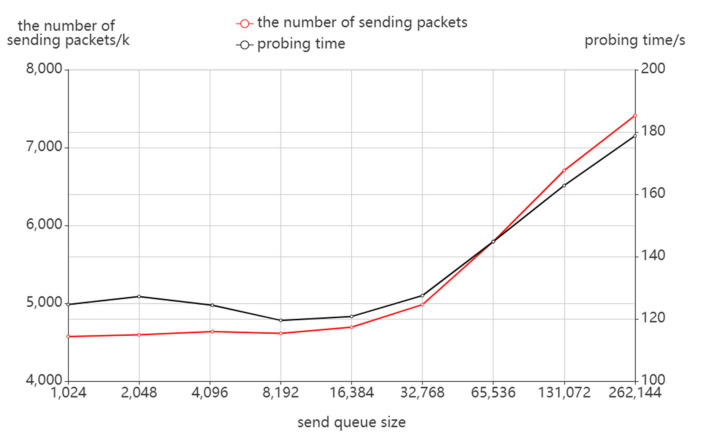
The effect of send queue size on packet sending.

**Figure 13 sensors-22-05650-f013:**
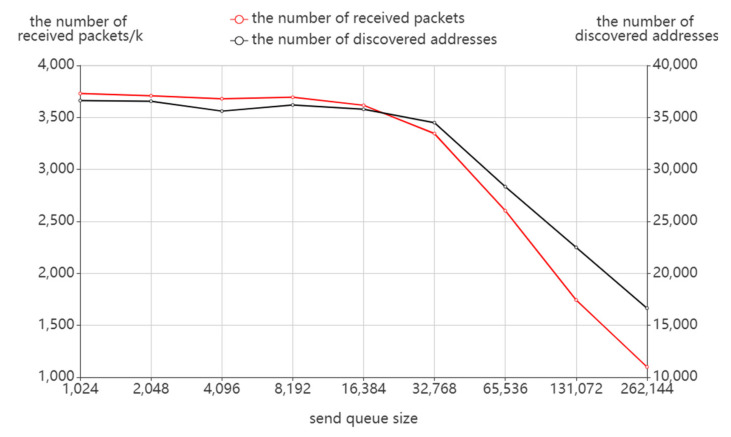
The effect of send queue size on packet receiving.

**Table 1 sensors-22-05650-t001:** TTL update policy at sending phase.

Condition		TTL Update Policy
curTTL≤initTTL	curTTL=1	curTTL=initTTL+1
	curTTL≠1	curTTL=curTTL−1
curTTL>initTTL	curTTL=maxTTL	Remove destination address
	curTTL≠maxTTL	curTTL=curTTL+1

**Table 2 sensors-22-05650-t002:** TTL update policy at receiving phase.

Condition		TTL Update Policy
TTL exceeded in transit	Destination address out of the stop set	Add the response address to the stop set
	rspTTL≤initTTL	curTTL=initTTL+1
ICMP echo reply	rspTTL>initTTL	Reset initTTL and curTTL
	rspTTL≤initTTL	maxTTL=rspTTL, Reset initTTL and curTTL
	rspTTL>initTTL	Remove destination address

**Table 3 sensors-22-05650-t003:** Comparison of different probing packets.

Packet	The Number of Sending Packages	The Number of Receiving Packages	Receiving and Sending Ratio	Probing Time (s)	DiscoveredAddresses
ICMP	5,019,079	1,577,417	31.43%	105.05	39,330
UDP	5,058,460	1,515,532	29.96%	99.423	39,121
TCP SYN	5,393,412	1,009,524	18.72%	102.188	38,310

**Table 4 sensors-22-05650-t004:** Comparison of large-scale path probing tools.

Approach	The Number of Sending Packages	The Number of Receiving Packages	Receiving and Sending Ratio	Probing Time (s)	DiscoveredAddresses
Yarrp	8,735,440	1,930,020	22.09%	249.04	37,990
FlashRoute	5,303,465	816,832	15.40%	113	39,610
Ours	4,608,352	1,565,637	33.97%	102.717	37,853

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
