# Peer review of "High-Speed Path Probing Method for Large-Scale Network"

_sensors, 2022, doi:10.3390/s22155650_

Round 1

Reviewer 1 Report

This paper can be accepted in current form

Author Response

Thank you for your valuable review comments.

Reviewer 2 Report

In this paper, the authors propose a large-scale network path probing approach to solve the problems of low probing efficiency and high probing redundancy commonly found in current research. However, there are some major concerns as follows:

 1. Clearly indicate the domain-independent innovative advance brought about by the proposed large-scale network path probing approach.

2. Variables need to be defined in detail, such as in Table. 1, authors need to give a detailed description of the variable curTTL, initTTL. In addition, the calculations involved in Table 1 suggest using formulas instead of text descriptions.

3. In the literature, there are some works on path probing in large-scale network. In recent years, deep learning technology has also been used to solve such work. Compared with those works, the difference and contributions of this work can be further highlighted. Some refs could be useful, e.g., (1) A privacy-protected intelligent crowdsourcing application of IoT based on the reinforcement learning[J]. Future Generation Computer Systems, 2022, 127: 56-69. (2) Design and analysis of probing route to defense sink-hole attacks for Internet of Things security. IEEE Transactions on Network Science and Engineering. At the authors' discretion.

4. The current text is read for attuned experts. The article is already long, but a few tutorial paragraphs on the techniques that make up the algorithm would make following the text easier for many readers. At the authors' discretion.

5. What are the limitations or restrictive assumptions behind the proposal?

6. The writing can be improved.

Round 2

Reviewer 2 Report

The authors have addressed my concern and it can be accepted now.

This manuscript is a resubmission of an earlier submission. The following is a list of the peer review reports and author responses from that submission.

Round 1

Reviewer 1 Report

Minor comments:

  - I recomend not to use Abbreviations in the Abstract. Define them the first time you write them in the regular text. Define al well (i.e.e bad defined is "Exponential moving average (EMA)") and once.

  - Please end the Section 1 with typical sentences about the structure of the paper.

  - It is important to define what IPID is.

Major comments:

  - The contributions you present are rather strange, because they are very related each other. I.e., the generation of addresses method? is part of the probing method? And, are they implemented in the "large-scale network path probing tool". You need to provide more clear contributions.

  - Why did you not use tipycal CAIDA tools? Please discuss.

  - Which is the effect on the network of your tool? Please discuss.

Reject Comments:

  - Please use a standard algorithmic language to define your algorithms.

  - Which were the formal verification techniques you used to show the validity of your algorithms. It is important you to clarify properly how did you implemented your algorithms providing typical UML architectural and implementation diagrams.

  - You mentionned that you used A chineese network to obtain your experimental results, and then you did worldwide tests. Plese explain better the nature of that measures (in chineese network and worlwide network).

  - Did you test the effect of typical ICMP filtering (rejecting them) packets in the network. How does this filters affect your mtehods.

Reviewer 2 Report

After carefully checking the paper, we found that this paper needs some modification and can be considered for publication in this journal.

Most of the given comments are to improve the paper's quality, and at the same time, they can gain more attention in the future.

The main idea should be clearer for the whole paper in many parts, like in the abstract section; the main idea and the author's contributions should be clearer as the main motivation behind this research.

The introduction section should present the main problem that has been solved and why the authors focus on this research on this problem. What is the primary proposed method here, and how does it solve the current problem?

The related works should at least be stated to focus on the main problems found in the literature, which can support the authors incoming this research.

The authors should focus on the proposed method and how to make it easy for the readers.

In general, update that lists by the following reference related to:

An Efficient 5G Data Plan Approach Based on Partially Distributed Mobility Architecture

Towards a better understanding of large-scale network models

The authors should added more related works as the number of used references is small

The paper's following can be clearer for the readers, and the structure can be changed in this term.

The results can be arranged better to highlight the main effect o the proposed method on the obtained results.

More details regarding the main procedure of the proposed method should be added into the Algorithm 2: Packet receiving algorithm.

The quality of the used figures is not ok. Improve them.

It is better to add more results in a tabular form.

Some figures seem not original, like figure 4.!!

The language should be improved as well

Round 2

Reviewer 1 Report

Minor comments:

  - I recomend not to use Abbreviations in the Abstract. Define them the first time you write them in the regular text. Define al well (i.e.e bad defined is "Exponential moving average (EMA)") and once.

Major comments:

  - The contributions you present are rather strange, because they are very related each other. I.e., the generation of addresses method? is part of the probing method? And, are they implemented in the "large-scale network path probing tool". You need to provide more clear contributions. IT IS CLEAR THAT A COMPARISON IS NOT A CONTRIBUTION. PLEASE RE-THINK THE CONTRIBUTIONS.

  - Which is the effect on the network of your tool? Please discuss. IT IS APPRECIATED OUR DISCUSION, BUT MORE EVIDENCES AND EXPLANATIONS ARE NEEDED SHOWING MORE THAN COMMENTS.

Reject Comments:

  - Please use a standard algorithmic language to define your algorithms. PLEASE FIX ERRORS LIKE "probing(the". SENTENCES LIKE "8 if not Check(packet) do 9 continue 10 end if" CAN BE IMPROVED CONSIDERABLY TO IMPROVE THE READABILITY OF THE ALGORITHM. YOU MUST DO AN EXTRA EFFORT TO SPECIFY BETTER THE ALGORITHMS.

  - Which were the formal verification techniques you used to show the validity of your algorithms. It is important you to clarify properly how did you implemented your algorithms providing typical UML architectural and implementation diagrams. IT IS TRUE WHAT YOU ANSWER BUT A FORMAL VERIFICATION OR A FORMAL MODEL OF YOUR METHOD IS NEEDED TO SHOW IT WORK IN GENERAL CASE. PLEASE RE-THINK.

  - You mentionned that you used A chineese network to obtain your experimental results, and then you did worldwide tests. Plese explain better the nature of that measures (in chineese network and worlwide network). MORE DETAILS ABOUT THE EXPERIMENTAL TEST ARE NEEDED.

  - Did you test the effect of typical ICMP filtering (rejecting them) packets in the network. How does this filters affect your mtehods. IT IS PROBABLY TRUE THAT YUR METHOD WORK BETTER THAN OTHER, BUT I AM NOT CONVINCED THAT IT PRODUCE GOOD RESULTS DUE TO ICMP FILTERING, AS YOU MENTIONNED.

Reviewer 2 Report

accept

Author Response

Thank you very much for your efforts and professional suggestions.

Round 3

Reviewer 1 Report

Minor comments:

  - I recomend not to use Abbreviations in the Abstract. Define them the first time you write them in the regular text. Define al well (i.e.e bad defined is "Exponential moving average (EMA)") and once. IPv4.

Major comments:

  - The contributions you present are rather strange, because they are very related each other. I.e., the generation of addresses method? is part of the probing method? And, are they implemented in the "large-scale network path probing tool". You need to provide more clear contributions. IT IS CLEAR THAT A COMPARISON IS NOT A CONTRIBUTION. PLEASE RE-THINK THE CONTRIBUTIONS.

  - Which is the effect on the network of your tool? Please discuss. IT IS APPRECIATED OUR DISCUSION, BUT MORE EVIDENCES AND EXPLANATIONS ARE NEEDED SHOWING MORE THAN COMMENTS. Two minutes of measures is not enougth, please re-consider experiments and formal demostrations.

Reject Comments:

  - Please use a standard algorithmic language to define your algorithms. PLEASE FIX ERRORS LIKE "probing(the". SENTENCES LIKE "8 if not Check(packet) do 9 continue 10 end if" CAN BE IMPROVED CONSIDERABLY TO IMPROVE THE READABILITY OF THE ALGORITHM. YOU MUST DO AN EXTRA EFFORT TO SPECIFY BETTER THE ALGORITHMS. Changes not enougth.

  - Which were the formal verification techniques you used to show the validity of your algorithms. It is important you to clarify properly how did you implemented your algorithms providing typical UML architectural and implementation diagrams. IT IS TRUE WHAT YOU ANSWER BUT A FORMAL VERIFICATION OR A FORMAL MODEL OF YOUR METHOD IS NEEDED TO SHOW IT WORK IN GENERAL CASE. PLEASE RE-THINK. The verification yuo commented is not presented well in the paper. Please Rethink.

  - You mentionned that you used A chineese network to obtain your experimental results, and then you did worldwide tests. Plese explain better the nature of that measures (in chineese network and worlwide network). MORE DETAILS ABOUT THE EXPERIMENTAL TEST ARE NEEDED. It is important to know abut the network to observe the efficiency of your method and scalability. Please re-think.

  - Did you test the effect of typical ICMP filtering (rejecting them) packets in the network. How does this filters affect your mtehods. IT IS PROBABLY TRUE THAT YUR METHOD WORK BETTER THAN OTHER, BUT I AM NOT CONVINCED THAT IT PRODUCE GOOD RESULTS DUE TO ICMP FILTERING, AS YOU MENTIONNED. Mix this answer with the answer due to the above question and please explain better these two aspects combined.
